# Blood pressure trends following birth in infants born under 25 weeks' gestational age: a retrospective cohort study

Emma Persad  ,[1,2] Björn Brindefalk,[3] Alexander Rakow[1,2]

[1]Department of Women's and Children's Health, Karolinska Institutet, Stockholm, Sweden
[2]Department of Neonatology, Karolinska University Hospital, Stockholm, Sweden
[3]Department of Molecular Biosciences, The Wenner-Gren Institute, Stockholm University, Stockholm, Sweden

**Correspondence to**
Dr Alexander Rakow; alexander.rakow@ki.se

## ABSTRACT

**Objective** The aim of our study was to describe postnatal blood pressure (BP) trends and evaluate relevant dynamics and outcomes for a subgroup of extremely preterm (EPT) infants.

**Design** Retrospective observational cohort study.

**Setting** Patients admitted to Karolinska University Hospital Stockholm.

**Patients** EPT infants born between 22+0 and 24+6 weeks' gestational age (GA) undergoing invasive, continuous BP monitoring through an umbilical arterial catheter.

**Main outcome measures** Physiological BP trends, the influence of cardiovascular active interventions and fluid boluses on BP, and relevant adverse outcomes, including intraventricular haemorrhage (IVH), necrotising enterocolitis (NEC) and death, were mapped over the first week of life.

**Results** We included 125 infants between January 2009 and November 2021. Mean BP values were 31 mm Hg, 32 mm Hg and 35 mm Hg, at 3 hours, 24 hours and 48 hours, respectively. A pronounced BP dip and nadir were observed around 20 hours, with a mean BP value of 32 mm Hg. 84% received fluid boluses within the first week of life; however, we could not observe any noteworthy change in BP following administration. Only 8% of patients received cardiovascular active drugs, which were too few to infer drug-specific effects. Overall, 48% developed IVH, 15% developed NEC and 25% died.

**Conclusions** Approximating clinically acceptable mean BP values using GA gives underestimations in these infants. The postnatal BP dip should be regarded as a physiological phenomenon and not automatic grounds for interventions which may momentarily stabilise BP but have no appreciable short-term or long-term effects. Further studies are warranted for improved understanding of clinically relevant trends and outcomes.

## INTRODUCTION

Despite increasing ability to care for extremely preterm infants (EPT), defined as those born under 28 weeks' gestational age (GA), considerable uncertainty surrounding age-appropriate and condition-appropriate blood pressure (BP) remains.[1–6] Furthermore, data on the efficacy of antihypotensive treatment and relevant thresholds to initiate treatment are lacking, despite episodes of

---

### WHAT IS ALREADY KNOWN ON THIS TOPIC

⇒ Low blood pressure (BP) affects 15–50% of extremely preterm (EPT) infants following birth and is associated with adverse outcomes; however, high-fidelity granular data outlining acceptable BP trends and accurate BP estimation practices crucial for comprehensive patient monitoring are lacking in this cohort. Further, a postnatal BP dip has been observed in EPT infants although is not yet well described or widely acknowledged, warranting clarification on the magnitude and timing of this dip and a high-quality descriptive analysis of BP to guide clinical care.

### WHAT THIS STUDY ADDS

⇒ We provide valuable insight into BP trends observed in a typical group of 125 EPT infants born at the border of viability (22+0 to 24+6 weeks' gestational age (GA)), finding a significant drop in BP and a nadir reached at approximately 20 hours after birth, regardless of cardiovascular active drug and fluid bolus administration. Additionally, we show that methods for predicting BP based on GA significantly underestimate the actual BP levels in this cohort.

### HOW THIS STUDY MIGHT AFFECT RESEARCH, PRACTICE OR POLICY

⇒ Clinical guidelines and practitioners should recognise the postnatal BP dip as a physiological occurrence, avoid GA-based BP estimation practices, acknowledge daily increments of BP and strive to use additional indicators of perfusion to access haemodynamic stability during assumed hypotensive episodes.

---

assumed hypotension often routinely being targeted with cardiovascular active drugs or volume boluses potentially associated with increased morbidity.[7 8] Comparatively, not intervening during hypotensive episodes can be equally harmful, particularly for neurodevelopment.[9–13] Clinical management of these infants is further complicated by their rapidly evolving physiology and numerous medical interventions, which may influence or mask natural BP trends.[14]

Neonatal intensive care units care for an increasing number of EPT infants prone to severe outcomes and complications, such as

intraventricular haemorrhage (IVH), necrotising entero-colitis (NEC) and early neonatal death.[4] [5] Although BP values for infants aged 23–25 weeks' GA have been reported in the literature, there are limited high-fidelity granular BP data on these infants, particularly as very few centres have access to comprehensive databases and invasively measured continuous monitoring data.[13]

A postnatal BP dip and immaturity-related BP fluctuations have been observed in EPT infants, although a comprehensive exploration of these phenomena and their long-term implications is not well described.[3] [15] Despite these BP variations being reported in the literature, clinicians commonly follow the consensus-based practice of estimating acceptable mean BP using the infants' GA in weeks.[13] [16] [17] Hypotension in EPT infants has also been defined as mean arterial pressure <30 mm Hg or less than the 10th percentile of gestational and postnatal age from published norms, which are often based on non-invasive measurements prone to overestimations and artefacts.[2] [3] [18] As providing evidence-based, high-quality care is crucial for these vulnerable infants and assumed hypotensive episodes often prompt invasive interventions to stabilise BP, clarification on age-appropriate BP and the magnitude and timing of the postnatal dip is warranted for improved clinical management. This study aims to describe BP in EPT infants born before 25 weeks' GA in our single-centre cohort, providing insight into the physiological development and outcomes of a typical group of vulnerable, underdeveloped patients subject to multiple supportive interventions.

## METHODS

This was a retrospective observational cohort study on EPT infants admitted to Karolinska University Hospital Stockholm between January 2009 and November 2021. We included all infants born between 22+0 and 24+6 weeks' GA with an umbilical arterial catheter enabling continuous invasive BP monitoring recorded at 1 min intervals. Infants over 25+0 weeks' GA, without invasive monitoring and with chromosomal abnormalities, were excluded (online supplemental figure 1).

Patient Monitoring Solutions (Clinisoft GE) was used to identify eligible patients using both GA and Diagnosis Related Group codes and retrieve monitoring data for all patients to limit selection bias. Patient documentation systems (TakeCare and Obstetrix) were used to extract demographic information and outcomes. All non-anonymised data were stored on Region Stockholm servers and not shared externally.

Data were extracted for BP values (systolic, mean, diastolic), time between birth and monitoring, incidence and time of IVH, NEC and death, and demographic factors, including GA (weeks+days), time of birth, weight (g), vasopressor or inotropic drugs (dopamine and dobutamine in this cohort), fluid boluses (fresh frozen plasma and sodium chloride), respiratory support, mode of delivery, maternal steroids and maternal chorioamnionitis. Our institution follows a protocolised treatment pathway for the provision of hypotension medication.[19]

We normalised the time elapsed from birth until start of invasive measurement, aggregating data on a per-hour basis to account for uneven sampling (in-house Perl script available on request). We focused our investigation between hours 1 and 168 of life, corresponding to a maximum of 76 million per-minute measurements, resulting in 16 555, 16 390 and 16 354 data points following IQR-based outlier removal for systolic, mean and diastolic BP values, respectively (online supplemental figure 2).

All statistical testing and data handling apart from initial extraction were done with R Statistical Software (V.4.0.3).[20] We evaluated statistically significant differences in time series using Granger's causality tests with bidirectional significance indicating similarity between curves. Modelling of significant factors for IVH severity was done with a generalised linear model (GLM) where numerical variables were centred and rescaled and other variables treated as factors (model: IVH grade~week+inotrope/vasoactive drug+NEC+>46 mm Hg at any time+%-time spent at >46 mm Hg+nr.fluctuations). For further analysis details, see online supplemental file 1. We did

**Table 1** Patient demographics, parameters and outcomes

| Demographics (125 infants) | n/data (%) |
|---|---|
| 22 weeks' GA | 11 (9) |
| 23 weeks' GA | 60 (48) |
| 24 weeks' GA | 54 (43) |
| Average weight | 627 g |
| Average time to invasive BP monitoring | 222 min–3 hours 42 min |
| HFOV received | 116 (93) |
| Cardiovascular active drugs received within 1 week | 10 (8) |
| Fluid bolus received within 1 week | 105 (84) |
| IVH any | 60 (48) |
| IVH grade I+II/III+IV | 45 (75)/15 (25) |
| NEC | 19 (15) |
| NEC operated | 18/19 (95) |
| Death overall | 31 (25) |
| Death within 2 weeks | 13 (10) |
| Average age at death overall | 35 days |
| Average age at death if died within 2 weeks | 5 days |
| Vaginal delivery | 70 (56) |
| Mother given antenatal steroids | 108 (86) |
| Chorioamnionitis present | 31 (25) |

BP, blood pressure; GA, gestational age; HFOV, high-frequency oscillatory ventilation; IVH, intraventricular haemorrhage; NEC, necrotising enterocolitis.

**Table 2** Average systolic (S), mean (M) and diastolic (D) blood pressure (mm Hg) and incremental change data for all 125 infants over time

| Time | Blood pressure (mm Hg) | | |
|---|---|---|---|
| | S | M | D |
| 3 hours | 39 | 31 | 24 |
| 12 hours | 40 | 33 | 26 |
| 24 hours | 42 | 32* | 25* |
| 36 hours | 44 | 34 | 26 |
| 48 hours | 46 | 35 | 27 |
| Value at nadir (time) | 41 (20 hours) | 32 (20 hours) | 25 (21 hours) |
| Increase from 3 hours to 12 hours | 1 | 1 | 2 |
| Increase from 3 hours to 24 hours | 3 | 1 | 1* |
| Increase from 3 hours to 36 hours | 6 | 3 | 2 |
| Increase from 3 hours to 48 hours | 7 | 4 | 2 |
| Change between 3 hours and 24 hours | 3 | 1 | 1 |
| Change between 24 hours and 48 hours | 4 | 3 | 1 |

*Indicates a decrease in mm Hg between 12 hours and 24 hours.

not involve patients or the public in the design, conduct, reporting or dissemination plans.

## RESULTS

### Demographics

This study included 125 infants born under 25 weeks' GA at Karolinska University Hospital Stockholm between 2009 and 2021. Our cohort comprised 11, 60 and 54 infants born at weeks 22, 23 and 24, respectively. Table 1 outlines baseline demographic data.

### BP trends during the first 48 hours

Increasing BP trends were observed over time. Our cohort had a mean BP of 31 mm Hg at baseline (3 hours after birth), 32 mm Hg at 24 hours (1 mm Hg increase from baseline) and 35 mm Hg at 48 hours (4 mm Hg increase from baseline). The BP nadir occurred at approximately 20 hours, with a mean BP of 32 mm Hg. Average systolic, mean and diastolic BP values and incremental changes from baseline are presented in table 2.

Figure 1A illustrates mean BP trends for all infants with the 25th and 75th quartiles, showing an initial increase and subsequent decrease during the first 24 hours. Figure 1B presents simplified findings suitable for clinical use.

Figure 2 illustrates BP trends for all infants during the first 48 hours with a pronounced dip and nadir observed at approximately 20 hours following birth.

Figure 3 illustrates the timing of the nadir for systolic, mean and diastolic BP values grouped by GA week.

### Differences in BP trends for various conditions

Granger's continuity tests showed significantly different BP trends when considering aggregate data for most conditions (online supplemental table 1). A refined representation of BP trends for infants in the three GA cohorts and plots comparing infants who died versus survived and developed IVH or NEC versus did not during the first 168 hours of life are shown in figure 4 and online supplemental table 3 (figure 4; online supplemental file 1).

### Model analysis and correlation of various factors with IVH

Using a GLM, we tested time spent in mean BP bands >46 mm Hg or <23 mm Hg to evaluate potential correlations with IVH grade as reported by Vesoulis et al, however found no such correlations (online supplemental table 2).[15] Interestingly, we found a significant correlation between infants exceeding a mean BP of 46 mm Hg at least once and the appearance of IVH at higher grades (p=0.02). To evaluate BP fluctuations, we calculated the number of the times BP crossed its own 24-hour moving average for a numerical measurement of BP stability, finding that a slow response to divergent BP was strongly correlated with the appearance of IVH at higher grades (p≤0.001). We also observed a statistically significant correlation between severity of IVH grade and lower GA (p=0.03) (online supplemental figure 4).

## DISCUSSION

This study provides high-fidelity granular BP data in infants born between 22 and 25 weeks' GA, contributing further insight to a limited body of evidence on this extremely vulnerable population.[3] Between 2009 and 2021, the clinical management of EPT infants at our centre, comprising ventilation strategies, fluid and bolus management, inotropic use and tolerance of hypercapnia, saw minimal changes. This stability can be attributed to our centre's conservative approach, emphasising the use

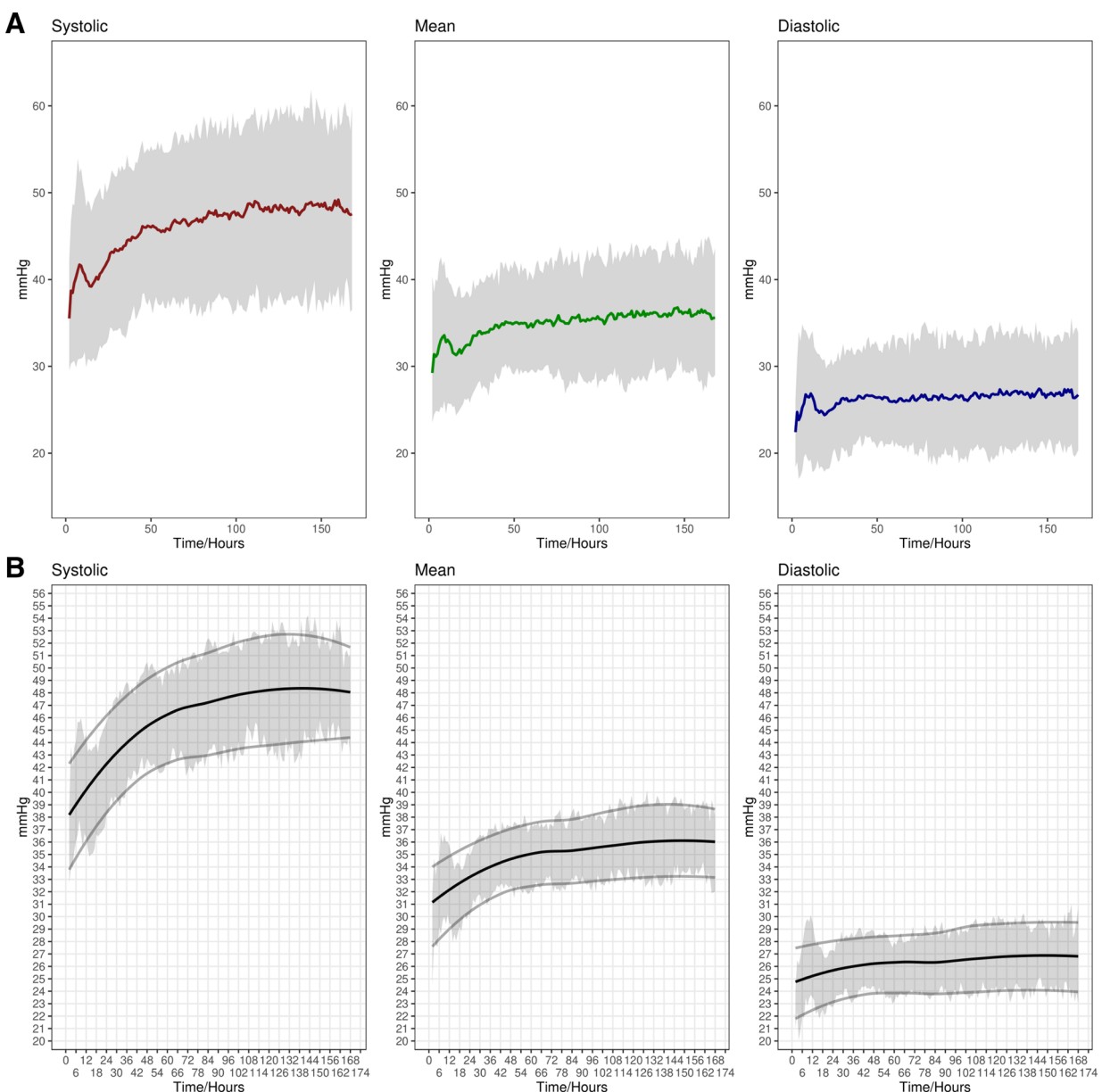

**Figure 1** (A) Presentation of total data for 125 infants; mean value and quartiles (shaded area) for systolic (red), mean (green) and diastolic (blue) values for hours 1–168. Shading corresponds to first and third quartiles for each dataset. (B) Plots showing the smoothed quintiles and mean value for systolic, mean and diastolic data for the first 168 hours, giving a simplified summary of expected blood pressure trends for extremely premature babies. Curves were smoothed with the Loess function and span=0.7 for easier interpretation.

of established and reliable methodologies. Notably, our ventilation strategy has remained unchanged since 2003, and the array of cardiovascular active drugs administered has not altered significantly. The only noteworthy modifications include the admission of infants born at 22 weeks' GA from 2016 and an increased emphasis on delayed cord clamping from 2018, meaning the overall management framework has adhered to a rigid, time-tested paradigm. As interventions to address suspected hypotensive BP can vary across countries and medical centres, our single-centre perspective provides a clear and consistent snapshot of our institution's practices and enhances the robustness of our findings.

Our cohort illustrated non-linear BP dynamics, finding a decrease in BP following birth with a nadir at approximately 20 hours (figures 2 and 3). We also observed significant differences in BP trends when comparing infants born at 22, 23 or 24 weeks' GA, highlighting the potential effect of immaturity on physiology, even for only a difference of 1–2 weeks.

Despite minimal GA differences between our cohorts, Batton *et al* described a considerably earlier decrease in BP, noting a nadir at only 4–6 hours following birth.[3] The transition from invasive to non-invasive measurement techniques was proposed to partly account for the observed decline in BP.[3] As our cohort exclusively received

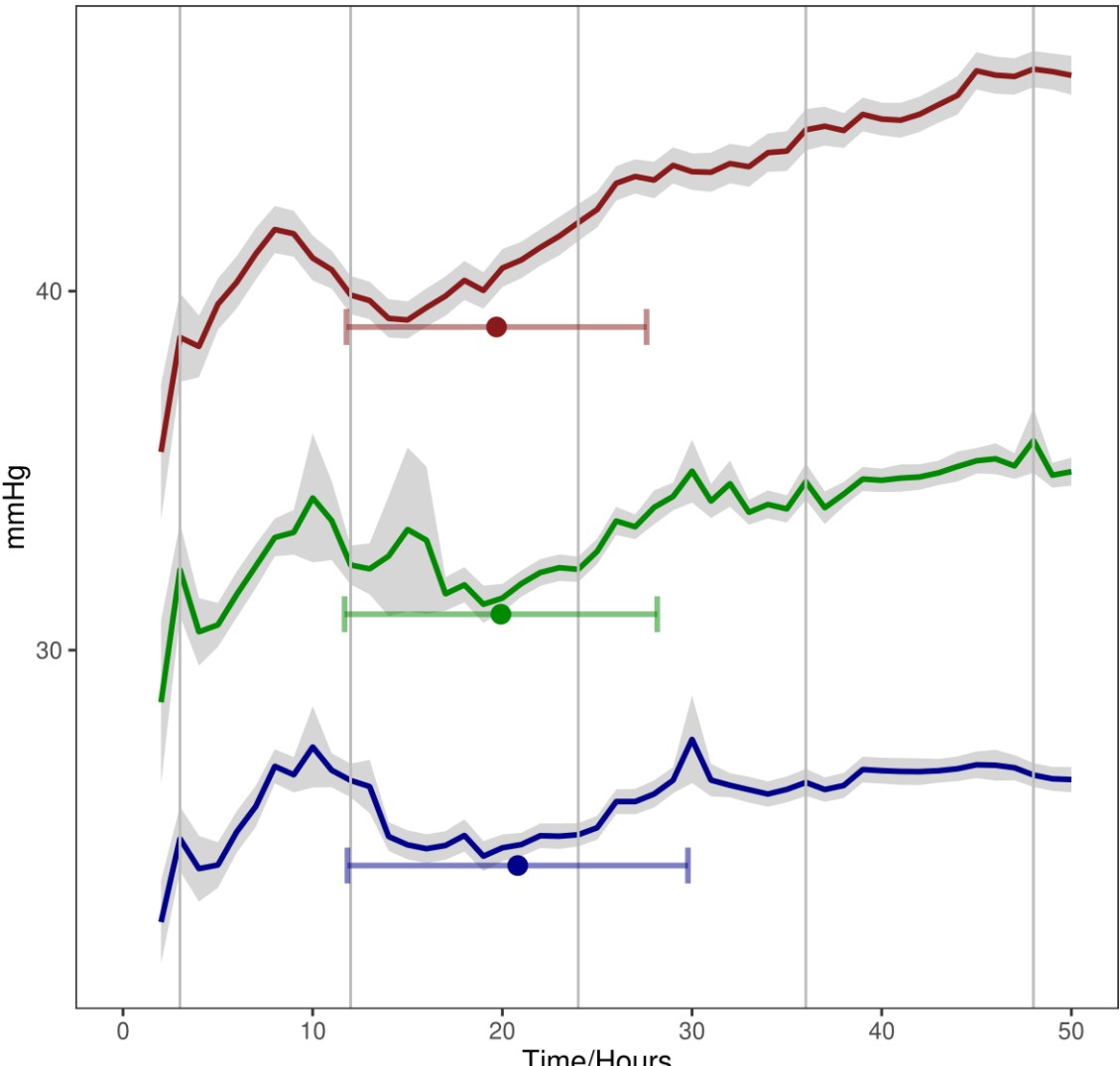

**Figure 2** Detailed plot showing blood pressure trends for all infants during the first 48 hours after birth with vertical lines corresponding to the time frames 3–12 hours, 12–24 hours, 24–36 hours and 36–48 hours. Shading corresponds to first and third quartiles for each dataset. The error bars indicate the mean nadir point and SD for the nadir point for each dataset.

invasive BP measurements, we consider the timing and dip in BP to be seemingly physiological and not attributable to measurement error, which is of immense clinical relevance as it often coincides with the initiation of antihypotensive treatment. Although the physiological mechanisms behind the dip remain elusive, Banerjee *et al* have suggested that a reduction in cardiac output may be a potential explanation.[21] Interestingly, they demonstrate a decrease in cardiac output with a nadir at approximately 12 hours primarily driven by a decrease in heart rate, which considerably precedes the BP drop seen in our cohort at 24 hours (online supplemental figure 3). We further investigated the relationship between heart rate and BP development, observing that BP began to recover approximately 10 hours before there was a corresponding increase in heart rate (online supplemental figure 3). Cardiac output has previously been shown to correlate poorly with mean BP, suggesting that other factors, such

as an increase in left-to-right shunt through a patent ductus arteriosus or changes in mean airway pressure due to the progression of respiratory distress syndrome (high-frequency oscillatory ventilation (HFOV)), should also be considered.[22 23] Further research into neuroendocrine factors, including catecholamines and hormones affecting systemic and pulmonary vascular resistance, should be prioritised to uncover the mechanisms underlying postnatal physiological development.[24]

Our findings add to the literature challenging the practice of GA-based estimation of postnatal BP in EPT infants.[15] In our cohort, infants born between 22+0 and 24+6 weeks' GA had a mean BP of 31 mm Hg at 3 hours, 32 mm Hg at 24 hours and 35 mm Hg at 48 hours, considerably higher than estimated using GA-based practices. Our findings were similar to Vesoulis *et al*, who estimated mean BP in infants <28 weeks' GA as 33 mm Hg following birth.[15] Similarly, Kent *et al* reported a mean

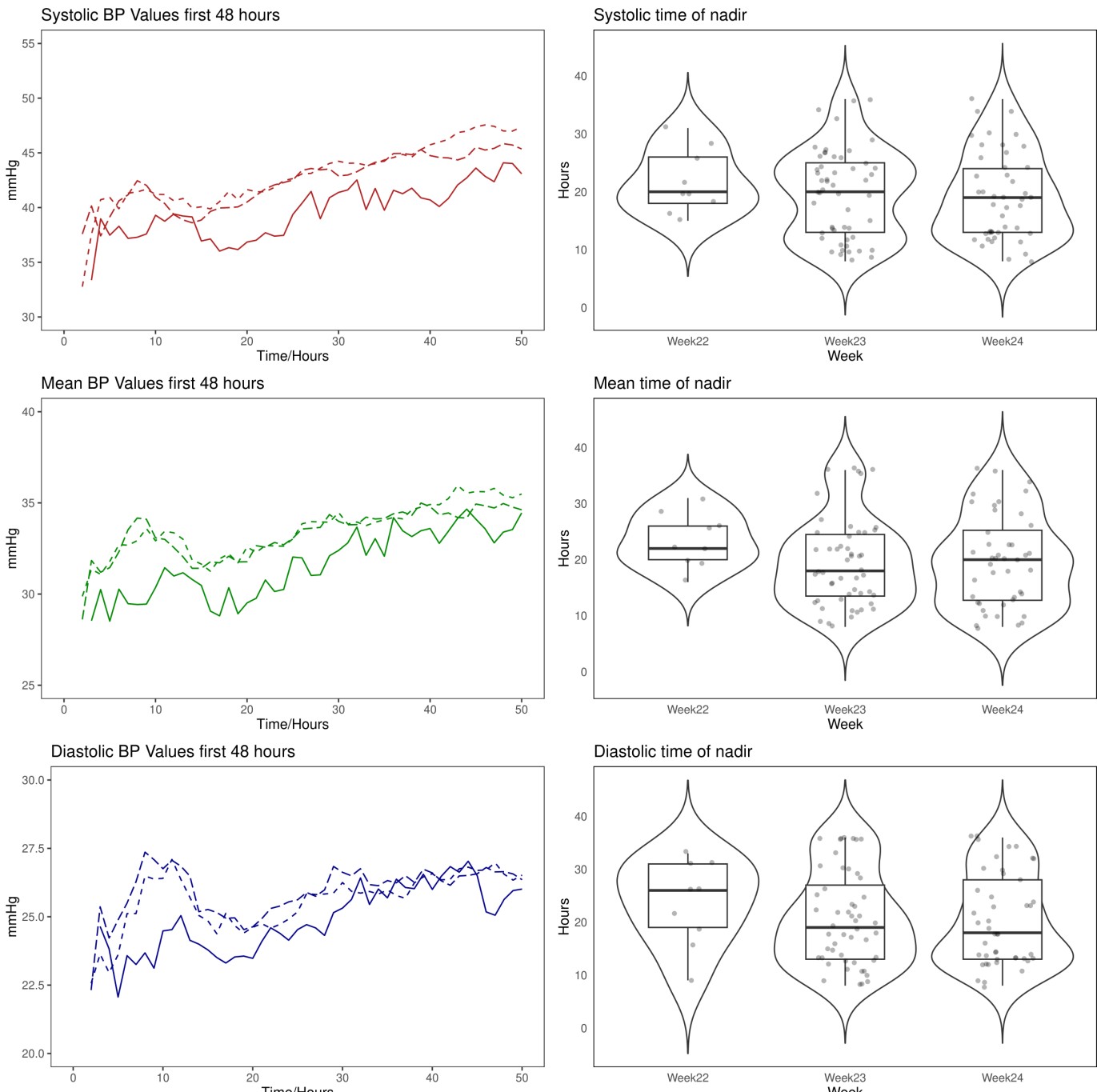

**Figure 3** Average blood pressure (BP) values (left panel column) and time of nadir point as determined on a per-patient basis during hours 8–36 (right panel column), illustrating the difference by gestational age week. Top row corresponds to systolic, middle row to mean and bottom row to diastolic BP. For left column, solid line corresponds to week 22, long dashed line to week 23 and short dashed line to week 24.

BP range of 31–43 mm Hg at 24 hours in infants aged 28–29 weeks' GA.[18] This encompasses our findings of 32 mm Hg, although their measurements were recorded non-invasively, potentially explaining their high upper range of 43 mm Hg. Similarly, Batton *et al* reported a BP increase of 0.2 mm Hg per hour, an increase which our cohort did not exhibit; however, they did not strictly use invasive measurement techniques, which remain the most reliable means of BP monitoring.[3]

Different mean BP trends were found between major comparisons, indicating that eventual death, IVH diagnosis alone, IVH and NEC diagnosis combined, GA, maternal factors and delivery method may be correlated with mean BP trends or influence mean BP enough to alter trends when compared with infants in the respective comparison groups (online supplemental table 1). However, these tests only revealed differences in trends and not one trend could be connected to specific

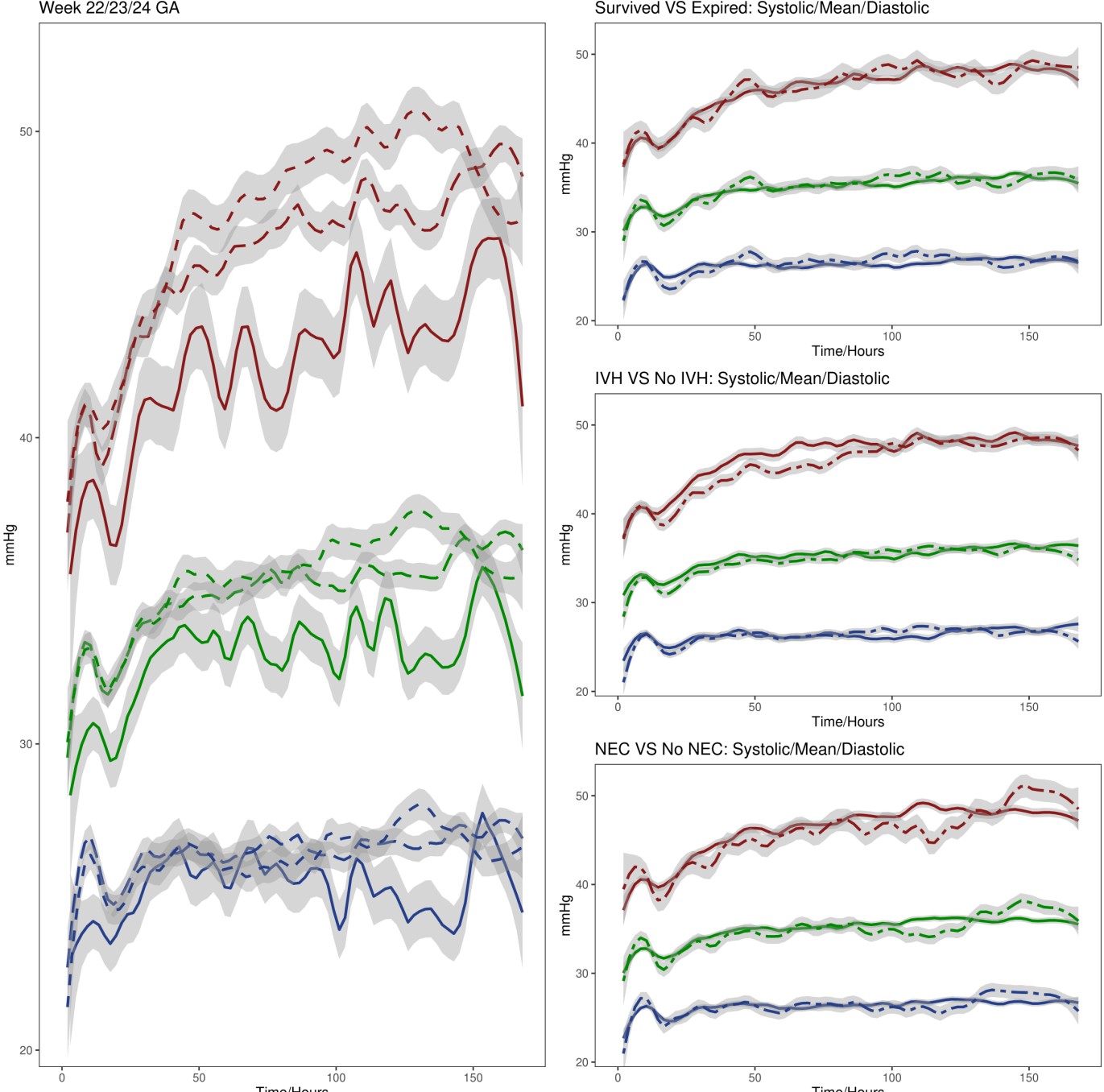

**Figure 4** Comparison of systolic (red), mean (green) and diastolic (blue) BP trends over time for different week cohorts. Solid lines correspond to week 22, long dashes week 23 and short dashes week 24 GA (left). Solid line indicates base condition (survived, no diagnosis), while dashed line indicates the alternative (right). Curves were smoothed with the Loess function and span=0.1. BP, blood pressure; GA, gestational age; IVH, intraventricular haemorrhage; NEC, necrotising enterocolitis.

outcomes, and the findings were only evident when considering aggregate data. Further exploration is required to draw more in-depth conclusions.

We found no correlation between time spent in specific mean BP bands suggested as thresholds (>46 mm Hg and <23 mm Hg) and eventual IVH diagnosis.[15] However, infants who had a mean BP that crossed 46 mm Hg at any time were significantly correlated with more severe IVH grade. Further, our data did not show the same correlation between increased instability and increased severity

of IVH observed by Vesoulis *et al*, but instead found that a slow response to divergent BP was strongly correlated with increased IVH severity. However, cohort differences and Vesoulis *et al* not detailing their statistical analysis rendered a direct comparison difficult.

While guidelines outlining thresholds for intervention would be clinically useful, our analyses highlight that the complexity of BP trends renders numerical values difficult to generalise and use. Retrospectively analysing BP trends over time may display patterns indicating a risk of

adverse outcomes; however, apart from having crossed a mean BP of 46 mm Hg, we could not identify stringent numerical limits to define clinically safe BP thresholds. Although clinicians are particularly inclined to intervene to stabilise BP in EPT infants, our data showed a seemingly natural dip following birth and no correlation between low BP and adverse outcomes.[15 18] If regarded as a natural physiological reaction, this dip should not be automatic grounds for systemic intervention, which may even be counterproductive considering the potential risks. Thus, it is inadequate to rely solely on BP to assess haemodynamic stability in clinical practice and imperative to use additional indicators of perfusion before commencing interventions.

This study provides quantitative and qualitative improvements to the literature, namely the high-fidelity granular data in a comparatively large infant cohort for this GA group. Our cohort also had a narrow GA range and very few received cardiovascular active drugs, allowing us to infer trends representative of this understudied population. However, our results were limited by only evaluating 11 infants born in week 22. Although this is novel for the literature, the findings for this subset should be viewed with caution. It is unclear whether the lesser fluctuations observed in this group are due to immaturity-related physiological factors or low patient numbers. Further, 93% of infants were on HFOV, as this is our first-line ventilation strategy for EPT infants, which has the potential to affect preload and afterload and thereby cardiac output, subsequently decreasing BP when high mean airway pressures are used. Only 8% of infants received cardiovascular active drugs during the first week of life, which we considered too few to investigate drug-related effects. 84% of our infants received fluid boluses or fresh frozen plasma during the first week of life; however, we could not observe any noteworthy change in BP following administration. We were also not able to obtain data on cord clamping procedures. Regardless, as it can be assumed that most infants in this age cohort receive the aforementioned interventions routinely from 2018, these results are still indicative of this population. Nevertheless, these values should be regarded as approximate guidelines when treating infants in this age range.

## CONCLUSION

This study presents highly accurate, invasive BP data on a typical cohort of unstable EPT infants born under 25 weeks' GA. Our findings highlight the postnatal BP dip as a likely physiological phenomenon of currently unclear aetiology which should not immediately prompt medical intervention provided other indicators of perfusion are favourable. Our results also challenge the GA-based practice of estimating mean BP, encouraging clinicians to reconsider this rule and acknowledge its vast underestimation in these infants. Overall, our data did not reveal specific BP trends to be correlated with poor outcomes, challenging the use of BP as a sole indicator of haemodynamic stability and warranting future investigations. Further research using invasive data on larger cohorts of EPT infants is necessary to clarify acceptable physiological BP trends to optimise clinical care.

**Acknowledgements** We would like to thank Raman Abram for his technical support and guidance with the patient databases.

**Contributors** EP made substantial contributions to the design and acquisition of data, drafted the article and revised it critically for important intellectual content, and approved the final version to be published. BB made substantial contributions to the analysis and interpretation of data, revised the article critically for important intellectual content and approved the final version to be published. AR made substantial contributions to the conception and design of the study, interpreted the data, drafted the article and revised it critically for important intellectual content, and approved the final version to be published. AR is the guarantor of this work.

**Funding** The authors have not declared a specific grant for this research from any funding agency in the public, commercial or not-for-profit sectors.

**Competing interests** None declared.

**Patient and public involvement** Patients and/or the public were not involved in the design, or conduct, or reporting, or dissemination plans of this research.

**Patient consent for publication** Not applicable.

**Ethics approval** This study involves human participants and was approved by the Swedish Ethical Review Authority (Etikprövningsmyndigheten; reference number: 2022-01588-01-262498). This was a retrospective cohort study using register data, so no consent was obtained or required.

**Provenance and peer review** Not commissioned; externally peer reviewed.

**Data availability statement** Data are available upon reasonable request. The data supporting the findings of this study (deidentified participant data) are available upon reasonable request to the corresponding author.

**ORCID iD**
Emma Persad http://orcid.org/0000-0003-2719-3685

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
