## [Reviewer comments · BMJ Paediatrics Open]

This paper was submitted to a another journal from BMJ but declined for publication following peer review. The authors addressed the reviewers' comments and submitted the revised paper to BMJ Paediatrics Open. The paper was subsequently accepted for publication at BMJ Paediatrics Open.

ARTICLE DETAILS

TITLE (PROVISIONAL)	Blood pressure trends following birth in infants born under 25 weeks' gestational age: a retrospective cohort study
AUTHORS	Persad, Emma Brindefalk, Björn Rakow, Alexander

VERSION 1 – REVIEW

REVIEWER	Dr. L Mahoney Yeovil District Hospital NHS Foundation Trust, Paediatrics
REVIEW RETURNED	16-Dec-2023

GENERAL COMMENTS

Thank you for the opportunity to review this manuscript. This is a single centre study conducted over 12 years which includes infants of a low gestational age and describes the blood pressure trends seen in these infants including a physiological postnatal dip. The manuscript does add important data to the existing literature but I do have some comments and questions outlined below.

Introduction:

I agree with the previous reviewer to which the authors have replied to. I do believe that there are some very important references that the authors do not include. For example the EpiPage2 study (PMID: 28302697) and Faust et al. PMID: 26199082. Whilst I understand a desire not to add more citations to a concise introduction I do believe that there are additional references which do describe blood pressure data in babies of a low gestational age or birthweight which would be important to include in the introduction.

Methods:

The authors mentioned about the use of various inotropes babies including cohort. Do they have any data on the use of fluid boluses over the time period that they measured blood pressure?

They outlined that babies only received either dopamine or dobutamine infusions. It is also mentioned in the discussion that "... the overall management framework has adhered to a rigid, time tested paradigm." I wondered as the infants in this cohort only received 2 different types of inotropes does the institute follow a protocolised treatment pathway for hypotension? Could the authors provide some detail on this in the methods section please.

	I note that in the discussion the authors mention that there was an increased emphasis on providing delayed cord clamping in the institution. Do they have data that they can provide on the percentage of infants that received delayed cord clamping? Results: The authors describe that the natural dip that they found in a infants blood pressure over the first 48 hours of life was presumably due to a reduction in cardiac output. I wondered whether they had compared the blood pressures between the babies who received dopamine versus dobutamine. As the latter is better at increasing cardiac output did they see the same physiological dip in dobutamine treated infants? Discussion: With regards to citation 18 by Banerjee at al. I would recommend that the authors update this to the full citation which has recently been published and supports their assumption that the blood pressure is caused by a possible reduction in cardiac output. I would also like to see the authors suggest the physiological mechanism as to why a drop-in cardiac output seen is linked to the physiological drop dip in blood pressure seen in this study. For example is it because of the inability of the preterm myocardium to deal with the increased afterload that we see due to the the increase in systemic vascular resistance that we see after birth? The authors maker statement "We also did not correlate data on cord clamping procedures or administration of fluid boluses and blood transfusions, however, it can be assumed most these in most instances in this age cohort receive these interventions routinely..." given this statement by the office I think it would be really important that if they do her data on cord clamping procedures, the administration of fluid boluses blood transfusions in this cohort that they provide it. As whilst they indicates that these routine interventions one could argue that in certain institutions that the administration of fluid boluses is done very cautiously in this cohort and would not be considered routine for example.
--	---

REVIEWER	Dr. Ebru Ergenekon Gazi University Hospital, Pediatrics
REVIEW RETURNED	26-Dec-2023

GENERAL COMMENTS	I find the study quite interesting with valuable data where the literature is rather limited. I have 2 comments; 1- The value of 46 mmHg is mentioned several times as a potential risky value for IVH however only in the supplement it is
--

	finally cleared as MEAN BP! So I think that should also appear in the main text that the critical 46 is the mean BP. 2- I think the reader should know when the vasoactive drugs or volume boluses were started/given. Do they correspond to the dip BP time or were they before. That would provide better understanding regarding the physiological condition.
--	--

REVIEWER	Jay Banergee Imperial College London
REVIEW RETURNED	27-Dec-2023

GENERAL COMMENTS	Very well written manuscript on a less well researched area lacking in good quality evidence. The authors have done a singlecentre retrospective observational study on blood pressure trends in babies born at 22, 23 and 24 weeks of gestational age over the first few days of life and have demonstrated the physiological changes that happen with a drop in the BP in the first few hours of life. In addition they have demonstrated that even a single BP recording above 46 mm Hg could result in development of IVH. Despite modernistic conservative approach to cardio active medications use around 40% of those who had medications developed IVH, indicating strong relationship between the use of these agents and IVH. I think the authors has proposed few very important messages as above which are important to be taken into account. I would suggest a couple of very minor comments:  1. Instead of using the word high-quality data in all places, they may want to use the word high fidelity granular data which is more descriptive. 2. The authors may want to change the reference 18 to a more recent one: 10.1136/archdischild-2023-325941 3. The authors may wish to think how the Supplementary Figure 1 could be included in the main manuscript as it demonstrates the dataflow. They may wish to swap one of the figures included to the main manuscript and put it in the supplement I strongly believe that this manuscript will be a great addition to literature in an area of limited evidence.
---

REVIEWER	Dr. Beau Batton Southern Illinois University School of Medicine
REVIEW RETURNED	31-Dec-2023

GENERAL COMMENTS

The inclusion 22 week GA infants is unique. Use of only UAC BP values adds strength to the investigation. Comments for consideration:

- 1) The Main outcome measures in the abstract are somewhat vague (e.g. "relevant adverse outcomes"). Some specificity may be helpful
- 2) The last 2 sentences of the Conclusion of the abstract are true but not really specific to this investigation. I would remove them and focus more on the results of this study
- 3) Introduction, page 5, lines 31 - 38 (last 2 sentences of 1st paragraph): References 3, 6, 9, 13, and Rabe. J Perinatol 2021 (DOI: 10.1038/s41372-021-01169-5) do include BP values for infants 23 - 25 weeks GA. The last sentence is speculative. Please delete it and edit the 2nd to last sentence.
- 4) Discussion, page 9, 1st sentence: This is not the first description of continuously invasive BP data in infants 23 - 25 weeks GA. It may be one of the few that reports only UAC BP data for only infants 23 - 25 weeks GA. However, other studies have looked at this sub-population (exclusively) and other similar BP graphs are available for this population. Please edit this sentence and update the reference list accordingly.
- 5) Only 11 infants born at 22 weeks GA were included over an ~13 year period. This is not enough to reach meaningful conclusions no matter how many data points those babies provided. I would not emphasize this as much as the current draft does.
- 6) The main finding this manuscript contributes to the existing literature is the decrease in BP observed from ~15-20 hours. I suggest highlighting this more with a slightly different approach. The 12 BP graphs in figures 1-2 plus the data in supplementary figures 2 - 4 plus the data in table 2 and the supplementary tables overwhelm the reader with so much data that the primary finding is somewhat obscured and lost. A retrospective study over a 13 year period can't meaningfully interpret the relationship between BP trends and infant outcomes. This is partly because if you are going to look at outcomes, then you need to include the entire patient cohort not just the patient with a UAC. The last 2 sections of the results along with Figure 3, Supplementary table 1 and 2, and Supplementary figure 3 do not add much to the existing literature and can only be interpreted with caution given the manuscript's limitations. I suggest removing them as well as the related portions of the Discussion. This would allow for greater Discussion of the (only) novel finding in the manuscript (decrease in BP from 15-20 hours) - see below.
- 7) The Abstract states a "pronounced BP dip and nadir was observed around 15-20 hours." That is evident graphically in 3 of the 6 figures in figure 1, figure 2, and the figures in figure 3. It is not seen in the bottom 3 smoothed figures of figure 1 or table 2 (difference of only 1 mmHg between 12 and 24 hours which is well within the standard error for any measure of BP). In addition, the observed decrease in BP isn't actually quantified anywhere in the text of the results. Supplementary Figure 4 attempts to provide additional relevant information but isn't mentioned in the text of the manuscript until the Discussion. This is the most important finding of the manuscript. Please explain it more and include

Supplementary Figure 4 as a figure in the main body of the

	manuscript. How much of a nadir was observed? Could differences in BP measurement, data collection, or data analysis potentially explain this finding? Including 3 smoothed figures in the 1st figure with BP graphs (Figure 1) which do not show a decrease in BP creates confusion for the reader. It would be helpful to explain the findings (and potential implications) of Supplementary Figure 4 to the reader.
--	--

VERSION 1 – AUTHOR RESPONSE

Reviewer: 1

Dr. L Mahoney, Yeovil District Hospital NHS Foundation Trust

Comments to the Author

Thank you for the opportunity to review this manuscript. This is a single centre study conducted over 12 years which includes infants of a low gestational age and describes the blood pressure trends seen in these infants including a physiological postnatal dip. The manuscript does add important data to the existing literature but I do have some comments and questions outlined below.

Introduction:

I agree with the previous reviewer to which the authors have replied to. I do believe that there are some very important references that the authors do not include. For example the EpiPage2 study (PMID: 28302697) and Faust et al. PMID: 26199082. Whilst I understand a desire not to add more citations to a concise introduction I do believe that there are additional references which do describe blood pressure data in babies of a low gestational age or birthweight which would be important to include in the introduction.

Authors: Thank you for your insight. We have looked into these references and added them into the introduction section accordingly.

Methods:

The authors mentioned about the use of various inotropes babies including cohort. Do they have any data on the use of fluid boluses over the time period that they measured blood pressure?

Authors: Thank you for your comments. We have since added information in the manuscript concerning fluid bolus administration (which we defined as NaCl boluses or provision of fresh frozen plasma). We investigated the administration of said boluses with changes in BP, however did not find any interesting effects of administration (figure below). We plan on investigating this further in an upcoming project with a larger cohort.

They outlined that babies only received either dopamine or dobutamine infusions. It is also mentioned in the discussion that "... the overall management framework has adhered to a rigid, time tested

paradigm." I wondered as the infants in this cohort only received 2 different types of inotropes does the institute follow a protocolised treatment pathway for hypotension? Could the authors provide some detail on this in the methods section please.

Authors: Our institution follows a protocolized treatment pathway. We have added information regarding this in the methods section.

I note that in the discussion the authors mention that there was an increased emphasis on providing delayed cord clamping in the institution. Do they have data that they can provide on the percentage of infants that received delayed cord clamping?

Authors: Thank you for your comment. Unfortunately, we do not have any data on delayed cord clamping procedure since its implementation, which is a practical problem inherent in retrospective studies due to not having all information readily available in the journaling system. We can assume that more infants receive it since 2016 and that the younger ones likely receive it for a shorter amount of time. However, we do not have data on the exact timing of this, so we have chosen not to speculate on this in the paper. We plan on looking into this in a future study which is also larger in scope and hope to provide more data on this. I hope you can understand.

Results:

The authors describe that the natural dip that they found in a infants blood pressure over the first 48 hours of life was presumably due to a reduction in cardiac output. I wondered whether they had compared the blood pressures between the babies who received dopamine versus dobutamine. As the latter is better at increasing cardiac output did they see the same physiological dip in dobutamine treated infants?

Authors: Thank you for your interesting comment. We conducted further analyses regarding cardiac output and have added some information in the discussion section. Important to note, we also investigated the exact timing of dopamine and dobutamine in our cohort, finding that far fewer infants actually received it during our exact observation period (the first week of life). For the patients we did have data for, the responses were highly divergent. As such, we did not conduct any further analyses comparing the two drugs as there were far too few patients to infer any considerable influence on BP.

Discussion:

With regards to citation 18 by Banerjee at al. I would recommend that the authors update this to the full citation which has recently been published and supports their assumption that the blood pressure is caused by a possible reduction in cardiac output.

Authors: The citation has now been updated.

I would also like to see the authors suggest the physiological mechanism as to why a drop-in cardiac output seen is linked to the physiological drop dip in blood pressure seen in this study. For example is it because of the inability of the preterm myocardium to deal with the increased afterload that we see due to the the increase in systemic vascular resistance that we see after birth?

Authors: Thank you for your interesting comment. We conducted further analyses regarding cardiac output and have added some information in the discussion section.

The authors maker statement "We also did not correlate data on cord clamping procedures or administration of fluid boluses and blood transfusions, however, it can be assumed most these in most instances in this age cohort receive these interventions routinely..." given this statement by the office I think it would be really important that if they do her data on cord clamping procedures, the administration of fluid boluses blood transfusions in this cohort that they provide it. As whilst they indicates that these routine interventions one could argue that in certain institutions that the administration of fluid boluses is done very cautiously in this cohort and would not be considered routine for example.

Authors: Thank you for your comment. Unfortunately, as said above, we do not have any data on delayed cord clamping procedure since its implementation. We also investigated the administration of boluses with changes in BP, however only noticed a non-synchronous dip unlikely to be solely related to BP. We plan on investigating this further in an upcoming project with a larger cohort. We will endeavour to extract any kind of cord clamping data we can get in this upcoming cohort.

Thank you for taking the time to review our manuscript.

Reviewer: 2

Dr. Ebru Ergenekon, Gazi University Hospital

Comments to the Author

I find the study quite interesting with valuable data where the literature is rather limited. I have 2 comments;

1- The value of 46 mmHg is mentioned several times as a potential risky value for IVH however only in the supplement it is finally cleared as MEAN BP! So I think that should also appear in the main text that the critical 46 is the mean BP.

Authors: Thank you for your comments and for catching our lack of clarity. We have since updated the manuscript to say mean BP.

2- I think the reader should know when the vasoactive drugs or volume boluses were started/given. Do they correspond to the dip BP time or were they before. That would provide better understanding regarding the physiological condition.

Authors: Thank you for your comment. We have since added information in the manuscript concerning timing of vasoactive drugs and fluid bolus administration (which we defined as NaCl boluses or provision of fresh frozen plasma). We investigated the administration of said boluses with changes in BP, however did not find any striking effect of administration, only a non-synchronous dip unlikely to be solely related to BP. We plan on investigating this further in an upcoming project with a larger cohort.

We would like to thank you for encouraging us to look into the timing of drug administration. Further examination of the timing of drug administration revealed a low number of patients receiving them during the study period. In light of this, we have decided to remove these results from the paper and present a more extensive analysis in an upcoming larger study. We hope that you can understand that it was not our intention to change some of these results so drastically, but we are happy with the discovery that our data is less influenced by drugs than previously thought.

Thank you for taking the time to review our manuscript.

Reviewer: 3

Jay Banerjee, Imperial College London

Comments to the Author

Very well written manuscript on a less well researched area lacking in good quality evidence. The authors have done a single-centre retrospective observational study on blood pressure trends in babies born at 22, 23 and 24 weeks of gestational age over the first few days of life and have demonstrated the physiological changes that happen with a drop in the BP in the first few hours of life. In addition they have demonstrated that even a single BP recording above 46 mm Hg could result in development of IVH. Despite modernistic conservative approach to cardio active medications use around 40% of those who had medications developed IVH, indicating strong relationship between the use of these agents and IVH.

I think the authors has proposed few very important messages as above which are important to be taken into account.

I would suggest a couple of very minor comments:

1. Instead of using the word high-quality data in all places, they may want to use the word high fidelity granular data which is more descriptive.

Authors: Thank you for your comments. We have adapted this wording accordingly.

2. The authors may want to change the reference 18 to a more recent one:
[10.1136/archdischild2023-325941](https://doi.org/10.1136/archdischild2023-325941)

Authors: We have updated the reference, thank you for providing it.

3. The authors may wish to think how the Supplementary Figure 1 could be included in the main manuscript as it demonstrates the dataflow. They may wish to swap one of the figures included to the main manuscript and put it in the supplement

I strongly believe that this manuscript will be a great addition to literature in an area of limited evidence.

Authors: Thank you for your insight. We received feedback from other reviewers to highlight the figure showing the dip in BP and have placed that figure in the main manuscript. However, we have decided to keep this figure still in supplementary material, as we don't want to overburden the reader with too many figures, especially now as one additional figure is added. We really appreciate that you think it is useful, though, and will consider having a similar figure in future publications investigating a larger cohort.

Thank you for taking the time to review our manuscript.

Reviewer: 4

Dr. Beau Batton, Southern Illinois University School of Medicine

Comments to the Author

The inclusion 22 week GA infants is unique. Use of only UAC BP values adds strength to the investigation. Comments for consideration:

1) The Main outcome measures in the abstract are somewhat vague (e.g. "relevant adverse outcomes"). Some specificity may be helpful

Authors: Thank you for your comment, we have adapted the abstract.

2) The last 2 sentences of the Conclusion of the abstract are true but not really specific to this investigation. I would remove them and focus more on the results of this study

Authors: Thank you for your comment. Overall, we looked into the points that you deemed important and adjusted the manuscript accordingly. We have rearranged the order and some of the wording of the sentences in the conclusion, however would still like to present the totality of our results, as we still consider them all to be answering the main aims we set out to answer. Hopefully that is understandable.

3) Introduction, page 5, lines 31 - 38 (last 2 sentences of 1st paragraph): References 3, 6, 9, 13, and Rabe. J Perinatol 2021 (DOI: 10.1038/s41372-021-01169-5) do include BP values for infants 23 - 25 weeks GA. The last sentence is speculative. Please delete it and edit the 2nd to last sentence.

Authors: Thank you for your comment, we have adapted the wording accordingly.

4) Discussion, page 9, 1st sentence: This is not the first description of continuously invasive BP data in infants 23 - 25 weeks GA. It may be one of the few that reports only UAC BP data for only infants 23 - 25 weeks GA. However, other studies have looked at this sub-population (exclusively) and other similar BP graphs are available for this population. Please edit this sentence and update the reference list accordingly.

Authors: Thank you for your comment, we have adapted the wording and reference accordingly.

5) Only 11 infants born at 22 weeks GA were included over an ~13 year period. This is not enough to reach meaningful conclusions no matter how many data points those babies provided. I would not emphasize this as much as the current draft does.

Authors: Thank you for your comment, we have adapted the wording accordingly.

6) The main finding this manuscript contributes to the existing literature is the decrease in BP observed from ~15-20 hours. I suggest highlighting this more with a slightly different approach. The 12 BP graphs in figures 1-2 plus the data in supplementary figures 2 - 4 plus the data in table 2 and the supplementary tables overwhelm the reader with so much data that the primary finding is somewhat obscured and lost. A retrospective study over a 13 year period can't meaningfully interpret the relationship between BP trends and infant outcomes. This is partly because if you are going to look at outcomes, then you need to include the entire patient cohort not just the patient with a UAC. The last 2 sections of the results along with Figure 3, Supplementary table 1 and 2, and

Supplementary figure 3 do not add much to the existing literature and can only be interpreted with caution given the manuscript's limitations. I suggest removing them as well as the related portions of the Discussion. This would allow for greater Discussion of the (only) novel finding in the manuscript (decrease in BP from 15-20 hours) - see below.

Authors: Thank you for your comment. We very much appreciate the recognition of the importance of the decrease in BP following birth and look forward to exploring it further in a larger cohort. Nevertheless, we still believe that our observations of BP physiology in this cohort add to the limited literature on BP monitoring in this cohort and place further emphasis on its investigation for future research. We look forward to expanding on the suggested figures with more patient data in follow up papers. As such, we have readjusted our figures, however have not removed them completely. We have also decided to keep our additional figures and tables in the supplementary information document, as we believe they help in keeping our analyses transparent and may be interesting to some readers. We hope you can understand.

7) The Abstract states a "pronounced BP dip and nadir was observed around 15-20 hours." That is evident graphically in 3 of the 6 figures in figure 1, figure 2, and the figures in figure 3. It is not seen in the bottom 3 smoothed figures of figure 1 or table 2 (difference of only 1 mmHg between 12 and 24 hours which is well within the standard error for any measure of BP). In addition, the observed decrease in BP isn't actually quantified anywhere in the text of the results. Supplementary Figure 4 attempts to provide additional relevant information but isn't mentioned in the text of the manuscript until the Discussion. This is the most important finding of the manuscript. Please explain it more and include Supplementary Figure 4 as a figure in the main body of the manuscript. How much of a nadir was observed? Could differences in BP measurement, data collection, or data analysis potentially explain this finding? Including 3 smoothed figures in the 1st figure with BP graphs (Figure 1) which do not show a decrease in BP creates confusion for the reader. It would be helpful to explain the findings (and potential implications) of Supplementary Figure 4 to the reader.

Authors: Thank you for your comment. We have added numerical data concerning the nadir in the Results section. We do not anticipate that differences in BP measurement, data collection, or data analysis could explain these findings, as all measurement and collection procedures are automatic following the placement of the UAC (received by 76% of our infants <25 weeks' GA). We have readjusted our figures, however have decided to keep the smoothed graph, as we find it to be interesting for clinical staff in our Swedish clinical setting. Although the dip is not clearly visible, we believe that our results highlight both BP physiology following birth alongside the dip, so we would like

to illustrate this physiological data. We have also decided to keep our additional figures and tables in the supplementary information document, as we believe they help in keeping our analyses transparent and may be interesting to some readers.

We would like to sincerely thank you for your feedback on this manuscript. As previously stated, we are excited about the findings regarding the dip and look forward to exploring this further in follow up studies in a larger cohort. We have taken your comments to heart and they will also influence how we conduct our research and make figures in the future.

Thank you for taking the time to review our manuscript.